

# Detecting sedimentation impacts to coral reefs resulting from dredging the Port of Miami, Florida USA

Margaret W. Miller[1,*], Jocelyn Karazsia[2,*], Carolyn E. Groves[1,3], Sean Griffin[4,5], Tom Moore[4], Pace Wilber[6] and Kurtis Gregg[2,5]

[1] Southeast Fisheries Science Center, NOAA-National Marine Fisheries Service, Miami, FL, United States
[2] Southeast Regional Office, NOAA National Marine Fisheries Service, West Palm Beach, FL, United States
[3] Rosenstiel School of Marine and Atmospheric Sciences, University of Miami, Miami, FL, United States
[4] Restoration Center, NOAA National Marine Fisheries Service, St. Petersburg, FL, United States
[5] Earth Resources Technology, Inc., Laurel, MD, United States
[6] Southeast Regional Office, NOAA National Marine Fisheries Service, Charleston, SC, United States
[*] These authors contributed equally to this work.

Corresponding author
Margaret W. Miller,
margaret.w.miller@noaa.gov

## ABSTRACT

The federal channel at Port of Miami, Florida, USA, was dredged between late 2013 and early 2015 to widen and deepen the channel. Due to the limited spatial extent of impact-assessment monitoring associated with the project, the extent of the dredging impacts on surrounding coral reefs has not been well quantified. Previously published remote sensing analyses, as well as agency and anecdotal reports suggest the most severe and largest area of sedimentation occurred on a coral reef feature referred to as the Inner Reef, particularly in the sector north of the channel. A confounding regional warm-water mass bleaching event followed by a coral disease outbreak during this same time frame made the assessment of dredging-related impacts to coral reefs adjacent to the federal channel difficult but still feasible. The current study sought to better understand the sedimentation impacts that occurred in the coral reef environment surrounding Port of Miami, to distinguish those impacts from other regional events or disturbances, and provide supplemental information on impact assessment that will inform discussions on compensatory mitigation requirements. To this end, in-water field assessments conducted after the completion of dredging and a time series analysis of tagged corals photographed pre-, during, and post-dredging, are used to discern dredging-related sedimentation impacts for the Inner Reef north. Results indicate increased sediment accumulation, severe in certain times and places, and an associated biological response (e.g., higher prevalence of partial mortality of corals) extended up to 700 m from the channel, whereas project-associated monitoring was limited to 50 m from the channel. These results can contribute to more realistic prediction of areas of indirect effect from dredging projects needed to accurately evaluate proposed projects and design appropriate compliance monitoring. Dredging projects near valuable and sensitive habitats subject to local and global stressors require monitoring methods capable of discerning non-dredging related impacts and adaptive management to ensure predicted and unpredicted project-related impacts are quantified. Anticipated increasing frequency and intensity of seasonal warming stress also suggests that manageable- but- unavoidable local stressors such as dredging should be partitioned from such seasonal thermal stress events.

## INTRODUCTION

Numerous dredging projects have resulted in widespread environmental effects on coral reef communities (e.g., *Dodge & Vaisnys, 1977*; *Bak, 1978*; *Rogers, 1990*; *Erftemeijer et al., 2012a*). Coastal dredging and port construction exacerbates sediment influx by resuspending benthic sediments (*PIANC, 2010*), and fine sediments tend to have greater effects on corals compared to coarse sediments (*Erftemeijer et al., 2012a*). The spatial extent of impacts from dredging can be variable, and in a severe case, water quality impacts have been detected up to 20 km away from the dredging activity when oceanographic features included unidirectional flow during the project (*Fisher et al., 2015*). *Erftemeijer et al. (2012a)* note poor understanding of the biological response of corals to sedimentation can result in inappropriate management of dredging projects and provide several examples of dredging operations near coral reefs where inadequate management contributed to significant damage to reefs and mortality of corals. However, establishing realistic and ecologically meaningful sedimentation thresholds, as permit conditions and for use as triggers in an adaptive monitoring and management program, can be a challenge in coral reef environments (*Erftemeijer et al., 2012a*). To effectively minimize negative impacts on corals and coral reefs, a combination of reactive (feedback) monitoring of water quality and coral health during dredging activities and spill-budget modelling of dredging plumes could be used to guide decisions on when to modify (or even suspend) dredging (*Erftemeijer et al., 2012a*).

The Port of Miami entrance channel traverses coral reefs within the northern portion of the Florida Reef Tract. Six coral reef or hardbottom features characterized by *Walker (2009)* surround the federal channel at the Port of Miami, and include the nearshore ridge complex, both north and south of the channel, Inner Reef north and south, and Outer Reef north and south (Fig. 1). The Inner Reef north[1] is composed of two reef habitat types, including a Ridge-shallow (western portion) and Linear Reef (eastern portion). The current direction in the outer sections is dominated by the Florida Current with strong north-northeasterly flows, but the flow environment is complex with common current reversals and multiscale vortices capable of transporting suspended material in complex patterns (*Martinez-Pedraja et al., 2004*; *McArthur, Stamates & Proni, 2006*).

The purpose of the Port of Miami expansion dredging project was to provide improved navigation and safety for larger vessels, including post-Panamax class ships. An Environmental Impact Statement (EIS) prepared by the Army Corps of Engineers (USACE) concluded the dredging would result in 13,355 m$^2$ (3.3 acres) of direct impacts (i.e., reef that was ground up and permanently removed) to Outer Reef north and Outer Reef south. The EIS concluded impacts may also include the resuspension and deposition of sediments on nearby coral reef assemblages, but the area of anticipated sedimentation impact was not quantified (*US Army Corps of Engineers, 2004*). Nine years later, the Port of Miami entrance channel expansion dredging project was implemented during a 17-month period between

[1] *Walker (2009)* refers to this coral reef feature as Inner Reef. This portion of the Florida Reef Tract lacks a Middle Reef, and USACE reports often misidentify the Inner Reef as Middle Reef or Reef 2.

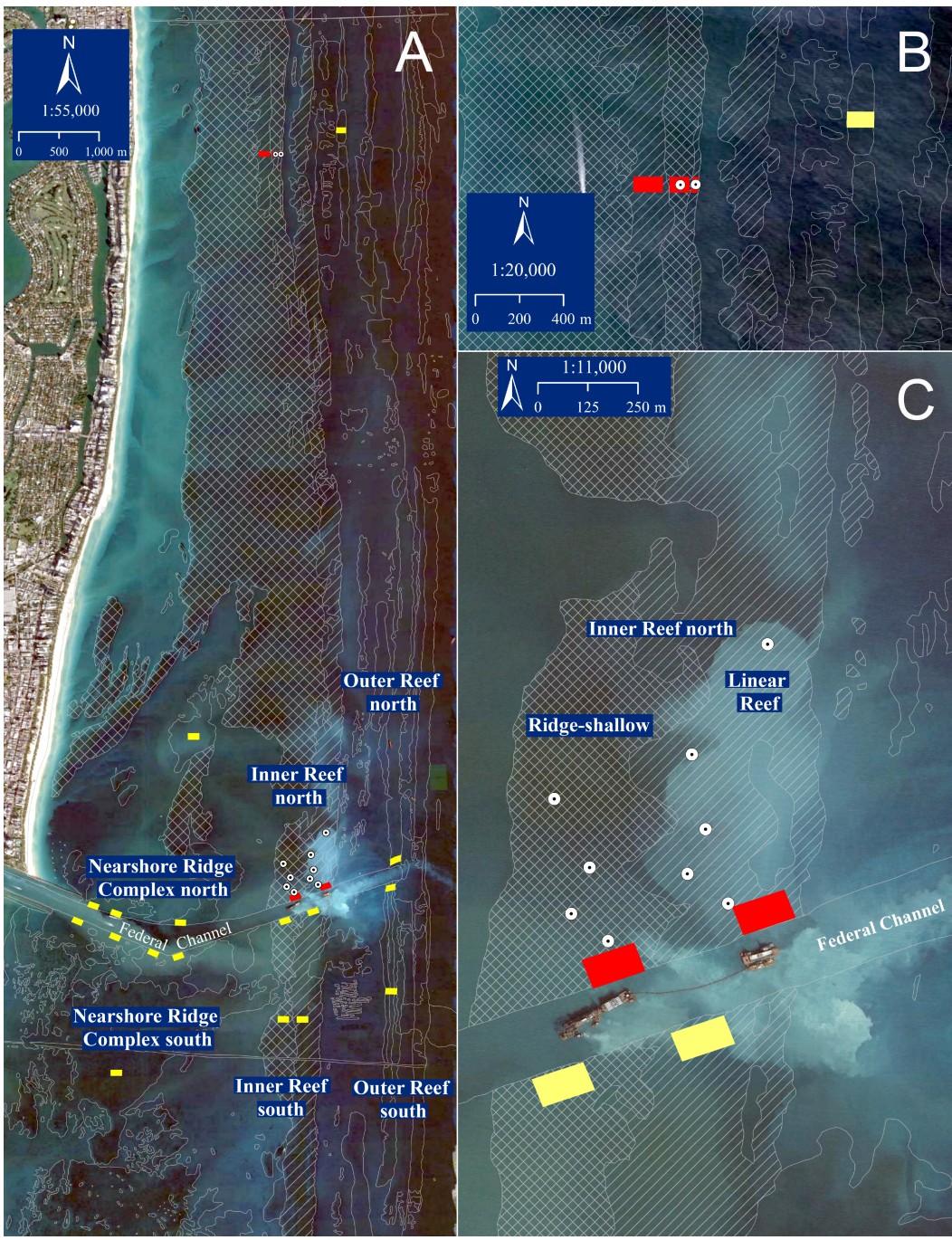

**Figure 1 Port of Miami study area.** (A) Port of Miami channel-side, sedimentation assessment and reference sites for all reef habitat types. Yellow boxes indicate locations of additional project-associated channel-side and reference sites for the Nearshore Ridge Complex, Outer Reef, and Inner Reef south that were not addressed in the current study. (B) Inner Reef north (red) and Outer Reef north (yellow; not addressed in the current study) reference sites, 9.3 km north of the channel. (C) Port of Miami channel-side and sedimentation assessment sites for the Inner Reef north. (continued on next page...)

**Figure 1 (...continued)**
Red boxes indicate locations of channel-side and reference area tagged coral colonies for Inner Reef north. Dots represent sedimentation assessment dive sites at 100, 200, 300, 500 m north of the channel in each habitat (Ridge-shallow depicted by cross-hatch, Linear reef by diagonal shading) of the Inner Reef, and 700 m north of the channel in the Inner Reef, Linear Reef. The base layer is from December 2014 (Google Earth Pro) during active dredging close to the Inner Reef north.

[2] Florida Department of Environmental Protection Permit #0305721-001-BI; ftp://ftp.dep.state.fl.us/pub/ENV-PRMT/dade/issued/0305721_Miami_Harbor_Phase_III_Federal_Dredging/001-BI/Final%20Order/Miami%20Harbor%20Final%20Order%2052212.pdf.

[3] Up to 40 days after the expansion dredging began.

November 20, 2013, and March 16, 2015 (Fig. S1). Additional maintenance dredging also occurred in the inner-harbor and federal channel, prior to and after the expansion dredging. A reported 4.39 million m$^3$ of material was dredged (CJ McArthur, US Environmental Protection Agency Region 4, pers. comm., 2016) via pipeline, backhoe, and clamshell dredges and deposited to a permitted offshore disposal site 2.4 km east-southeast of the project site in 120–240 m depth (*US Environmental Protection Agency, 1995*).

A permit[2] was issued in 2012 to the USACE by the Florida Department of Environmental Protection and included conditions for biological monitoring areas adjacent to the channel along each of six coral reef or hardbottom features (north and south of the channel in the Outer Reef, Inner Reef, and Nearshore Ridge; Fig. 1). All project-required monitoring stations were located within 50 m north and south of the channel (channel-side) in addition to reference areas located between 1.2 and 9.3 km from the channel (Fig. 1). The two baseline surveys were conducted in August 2010 (*US Army Corps of Engineers, 2011*) and October 23 through December 30, 2013[3] (*US Army Corps of Engineers, 2014*). While the former baseline assessment included stations up to 450 m north of the channel on the Inner Reef north, the baseline surveys from 2013, during-dredging, and post-construction monitoring included only potential impact locations within 50 m of the channel.

*Barnes et al. (2015)* undertook an independent remote sensing analysis partitioning natural drivers of sediment plumes near the Port of Miami channel, such as storms, wind or runoff events, from dredging-associated sediment plumes. They determined that sediment plumes detectable from satellite imagery during the dredging period were of 5× greater extent (127–228 km$^2$ during dredging compared to 18–46 km$^2$ under normal conditions) and 23–84% greater frequency than a baseline period prior to the start of dredging. This study also documented the greatest frequency and intensity of dredging-associated sediment plumes over the Inner Reef north reef sector. For this reason, we focused post-hoc sediment impact assessment effort in this reef sector, recognizing that additional reef area was likely impacted in the other sectors, with perhaps lesser intensity over a smaller extent.

The purpose of the current study was to better understand the sedimentation impacts that occurred in the coral reef environment surrounding Port of Miami, to distinguish those impacts from other regional events or disturbances, and provide supplemental information on impact assessment to inform discussions on compensatory mitigation requirements. This paper reports results from post-hoc field sampling focused on coral condition and standing sediment on reef substrates. Sampling was conducted at five sedimentation assessment locations within the Inner Reef north sector. Parallel sampling was conducted in the associated reference area, chosen and followed as part of the compliance monitoring program, approximately 9 km north of the channel and composed of similar reef habitat types (Fig. 1). In addition, analyses of photographic time series of individual tagged coral colonies within

channel-side and reference locations throughout the project provide a Before/After comparison for coral status over the project duration. Given the post-hoc nature of this analysis, we emphasized persistent effects, namely coral mortality or partial mortality related to sedimentation and persistent sediment presence in the reef environment. This conservative assessment does not address additional physiological stress and/or more temporary effect pathways likely also invoked by project associated turbidity and light attenuation (*Jones et al., 2016*) because post-hoc analysis using available data was not possible.

## METHODS

### Post-hoc field sampling

In December 2015, field sampling was conducted to quantify coral condition and standing sediment on reef substrates at locations spanning increasing distance from the channel in the Inner Reef north sector, in addition to the reference location (all 8–10 m depth; Fig. 1; Fig. S1). These 'sediment assessment' locations were spaced at 100, 200, 300, 500 and 700 m from the channel. At each distance except 700 m, transects were evenly distributed in both Ridge-shallow (RR) and Linear Reef (LR) habitat types. At the 700 m distance, only the Linear Reef habitat was assessed due to dive-time limitations. The reference location also included transects sampled within both Ridge-shallow and Linear Reef habitat. The project reference reefs were designated in a permit as part of the required compliance monitoring and were at least 0.8 km away from the channel (with the Inner Reef North reference sites located 9.3 km to the north). This distance was expected to be far enough away to prevent confounding effects from background channel turbidity, sedimentation, and effects from the commercial anchorage. The reference reefs were examined by divers and verified as representative of the intended habitat type (*US Army Corps of Engineers, 2010*).

Using Google Earth Pro, specific dive sites for each sediment assessment location were randomly selected to be at or near the 100-m interval mark and include a dive site in both Ridge-shallow and Linear Reef habitat type. Exceptions were three dive sites (100-RR, 200-RR, and 300-RR) that were selected to correspond with dive sites surveyed in a pre-construction assessment the USACE completed in 2010. Overall, this results in five sedimentation assessment locations (100, 200, 300, 500, and 700 m north of the channel), two dive sites per location—one in the Ridge-shallow and the other in the Linear Reef—with the exception of 700 m, where only the Linear Reef was surveyed. A temporary marker buoy was deployed from the boat at the pre-determined coordinates. At each dive site, two 50-m long transects were run in opposite directions from the buoy and sampled at 1.0-m intervals (50 point samples per transect, two transects per site). Observers recorded the occurrence of standing sediment along the line-intercept transects, where present, in two categories. 'Sediment-over-hardbottom' (SOHB) was designated if there was a visible accumulation of sediment. For example, algal turfs normally have some sediment embedded within them, but if the turfs were engulfed by sediment, this would be labelled as SOHB. If the sediment was qualitatively observed to be deeper (estimated >4.0 cm), it was labelled as "deep sediment over hardbottom" (DSOHB). In addition, at every 5.0 m along each transect, the depth of the sediment (cm) over hardbottom was measured with a ruler and the deepest of several measurements within one meter of the sample point was recorded.

Six 10 m² belt transects were also sampled along or parallel (within 10 m distance) with the two line-intercept transects at each dive site to quantify the condition of coral colonies within each area. Each scleractinian colony was recorded by species and condition; namely if the coral displayed recent partial mortality (i.e., minimally encrusted skeleton in which individual calyces were still discernable, *Lirman et al., 2014*), sediment present on live coral tissue (sediment accumulation), active disease (distinct white skeleton progressing across the colony), bleaching, "halo" mortality, or healthy if there were no noticeable signs of stress present. A "halo" refers to a pattern of partial colony mortality in which a concentric ring of dead coral skeleton occurs at the base of the coral colony as results from prior burial of the colony edges (Figs. S2A–S2C).

One-way ANOVAs followed by post-hoc tests between the assessment and reference locations were used to determine statistically significant differences in each survey parameter. For each of six parameters, (% cover SOHB, % cover DSOHB, sediment depth, and prevalences of recent partial mortality, 'halo' partial mortality, and sediment accumulation), preliminary one-way ANOVAs (on ranks, due to violation of parametric assumptions) showed no significant differences between the two habitat types ($p$-values ranging from 0.134 to 0.975 among the six parameters). Thus, transects of both habitat types were pooled at each location (i.e., distance from channel or reference) to increase replication and power to detect differences among the locations via one-way ANOVAs (on ranks when parametric assumptions were violated).

### Before/After analysis of coral status, qualitative and quantitative

Public records of time series photographs of tagged coral colonies taken as part of the compliance monitoring were obtained from the USACE. Tagged colonies were distributed in both the Ridge-shallow and Linear Reef habitat types at the channel-side and the reference (∼9.3 km north; Fig. 1) locations which had been designated for the permit monitoring. The colonies tagged at each location were of mixed species composition according to what was present, but included *Porites astreoides, Solenastrea bournoni, Pseudodiploria strigosa, Stephanocoenia intersepta, Meandrina meandrites, Siderastrea siderea,* and *Dichocoenia stokesii*. Photographs were taken at irregular intervals between a four-week pre-construction phase[4] (October–November 2013) and a four-week post-construction phase (July 2015). Intervals between images ranged from 1 to 121 (mean 14–18) days for the channel-side colonies and from 1 to 145 (mean 12–13) days at the reference site. The frequency of images varied according to permit requirements; photographs were to be taken at greater frequency when dredging was occurring in close proximity (<750 m) to the monitoring site. We thus made the conservative assumption that long intervals between photos corresponded to times when dredging activities were relatively distant and less influential, and that relevant instances of sediment interaction with the colonies are appropriately captured in the available time series for each colony. The time series of each colony was examined and the temporal sequence of conditions affecting each colony was noted. Specifically, the presence of sediment accumulation on live tissue, partial sediment burial generally of colony edges, complete colony burial by sediment, the presence of active White Plague disease signs (i.e., bright white exposed skeleton along colony margins, generally with a scalloped shape,

[4]A portion of the pre-construction phase overlapped the onset of expansion dredging (Fig. S1).

grading into gradually more encrusted, longer dead, skeleton), and 'sudden death' (the complete mortality of a colony between sequential photos in the time series, presumably attributable to disease, though no active disease signs were observable) were recorded in sequence. We included 'sudden death' in a category of disease impact given consistency with described patterns of mortality (i.e., complete colony mortality over a period of weeks) associated with a regional outbreak of 'White Plague' disease affecting most species of mounding corals during this time frame (*Precht et al., 2016*), a lack of other known disturbances such as storms, and the presumption that the longer interval of images during which most of the 'sudden death' occurred (winter 2014–2015) was during a period when dredging activities were distant according to permit requirements (hence transient sediment burial unlikely). The potential effect of sediment stress on susceptibility was examined by estimating the risk of subsequent disease and/or death in a group of colonies which had previously experienced partial sediment burial compared to the remaining colonies which had not shown partial burial in the time series photos. From this same set of time-series photographs, the live tissue area was quantified from the best-matched photo (angle and orientation) of each colony from the pre-construction and from the post-construction phase (generally four weekly photos in each phase) using the software CPCe (*Kohler & Gill, 2006*). Each photograph was calibrated using a scale bar with 5-cm increments included in each image, and then the area ($cm^2$) was calculated by outlining the live tissue area for each colony. Proportional change in live tissue area was calculated for each colony (i.e., (post-pre)/pre). Colonies which went missing prior to the post-construction phase were excluded from this analysis. The colonies from the two habitat types in each location were pooled and the change in colony area between locations was compared by a Mann–Whitney $U$-test.

## RESULTS

### Post-hoc field sampling

The mean percent cover of reef substrate characterized as "sediment over hardbottom" (SOHB) and "deep sediment over hardbottom" (DSOHB) was higher along the Inner Reef north transects (incorporating both Ridge-shallow and Linear Reef habitat), than the reference location transects (within the same habitat strata; Fig. 2A). The mean percent cover of SOHB was 17.5 to 36.0× higher at Inner Reef north locations, when compared to reference location (Fig. 2A), representing significant differences between each Inner Reef north location and the reference (One- way ANOVA $p = 0.002$ followed by post-hoc Holm-Sidak comparisons of each location with the reference at $p < 0.05$). At the reference location, 1% of the survey points exhibited reef substrate characterized as SOHB. The mean percent cover of DSOHB showed significant variation among sampled locations (one-way ANOVA on ranks, $p = 0.045$), ranging from 0.8 to 10.8% at distances 100, 200, 500, and 700 m from the channel (Fig. 2A), compared to no DSOHB recorded at the reference location nor the location 300 m north of the channel. However, statistical power was not adequate to discern significant differences for DSOHB among locations.

The mean depth of sediment was significantly higher, ranging from 2.7 to 10.0× higher, at Inner Reef north locations (Fig. 2B), compared to that measured at the reference location

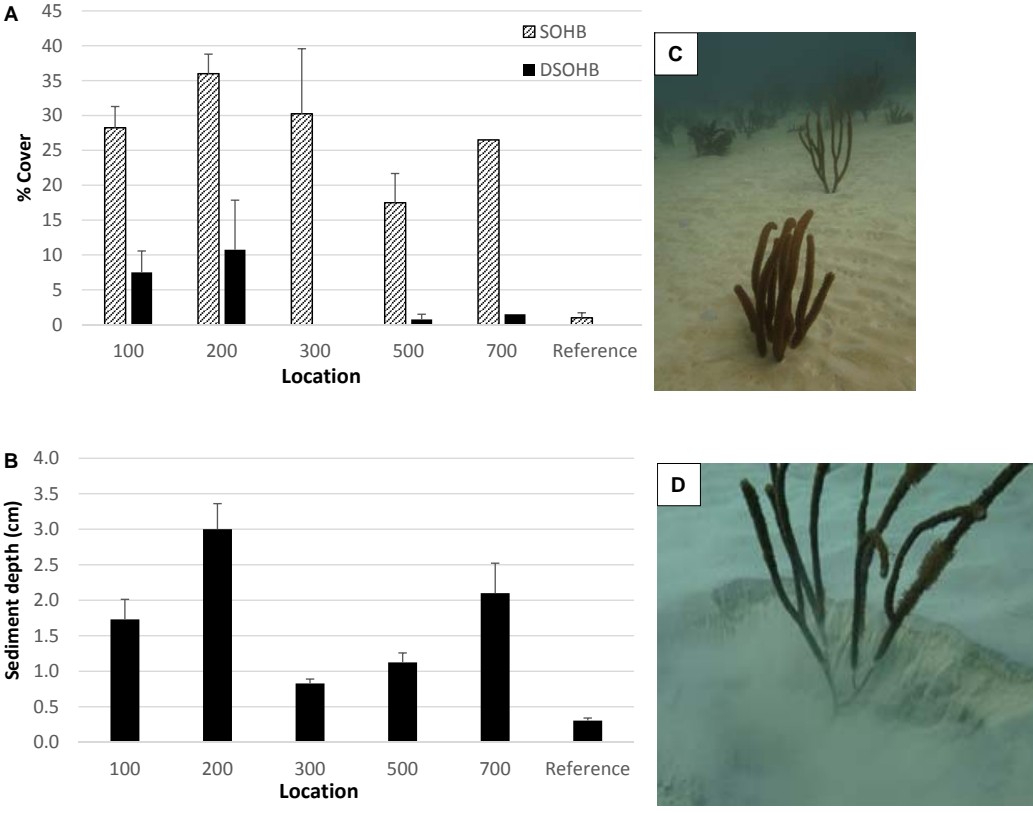

**Figure 2 Sediment cover on reef substrate.** (A) Mean (±1 SE) percent cover of sediment over hardbottom (SOHB) and deep sediment over hardbottom (DSOHB; >4 cm depth) along line point-intercept transects at sites of increasing distance from the channel and reference site. $N = 4$ transects for each, except 700 m where only two transects were sampled (hence no error bars are given). (B) Mean (±1 SE) depth of sediment at 0.5 m intervals along the same transects. Both habitat types were sampled at all sites except 700 m (LR only). Each of the sediment assessment locations had significantly higher SOHB cover and sediment depth than the reference area in post-hoc comparisons following one-way ANOVAs. (C–D) Illustration of expanse of deep sediment at the 200 m location showing soft corals with several cm burial (photos taken 11 Dec 2015; 25.763852°N, 80.098928°W).

(one-way ANOVA on ranks $p = 0.001$ followed by Dunn's post-hoc comparisons with reference). Specifically, along the transects located 200 m north of the channel, the mean sediment depth was 3.0 cm compared to 0.3 cm at the reference location (Figs. 2B–2D).

There was up to a 3.1 to 5.1× increase in the prevalence of corals with recent partial mortality at sediment assessment locations when compared to reference (Fig. 3; one-way ANOVA on ranks $p = 0.009$). Specifically, the 100, 300, and 700 m locations were significantly different than the reference (Dunns' post-hoc comparisons with reference, $p < 0.05$). The occurrence of sediment accumulation (SA) on live coral tissue ranged from 4.8 to 21.3× higher at sedimentation assessment locations when compared to the reference location with the 100 m and 200 m locations being statistically higher (SA; one-way ANOVA on ranks $p = 0.002$ followed by Dunn's post-hoc comparisons with reference; Fig. 3). Sediment halos (mortality at the base of colonies due to elevated levels of sedimentation) on scleractinian corals ranged from 3 to 26× more frequent at

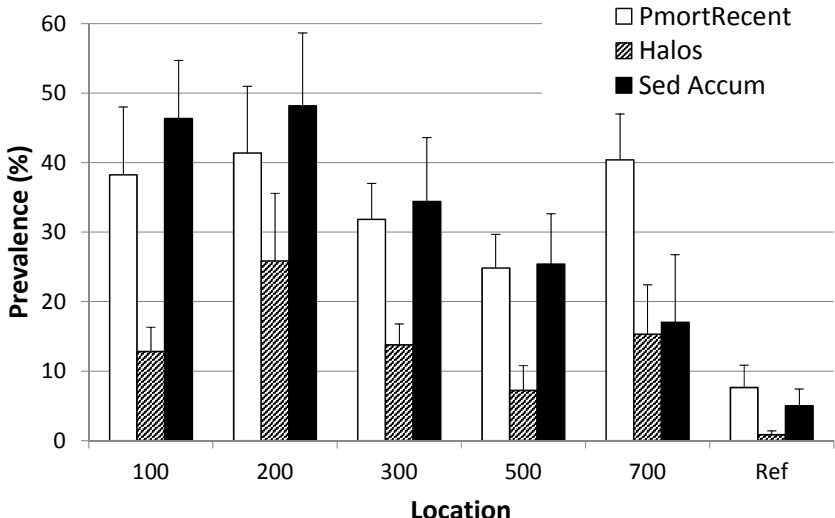

**Figure 3 Coral conditions.** The overall prevalence (mean + 1 SE) of colony conditions at sites spanning a gradient of distance from the dredged channel (100–700 m) and the reference site. Habitat types are pooled ($n = 12$ transects per location) with exception of 700 m site (only Linear Reef habitat sampled, $n = 6$ transects) and 500 m and 300 m ($n = 13$ transects per location, with the one additional transect being in the Linear Reef habitat). Sed Accum, sediment presence on living coral tissue; PmortRecent, recent partial mortality among colonies (*Lirman et al., 2014*); Halos, distinct pattern of partial mortality (not necessarily recent) in which tissue loss manifests as an outer concentric ring or partial ring which is consistent with that resulting from previous partial burial of the colony (see Fig. S2 for illustration). Ref, Reference location.

sedimentation assessment locations when compared to the reference location with the 200 m and 300 m locations being significantly higher (Fig. 3; one-way ANOVA on ranks $p = 0.011$ followed by Dunn's post-hoc comparisons with reference).

## Before/After analysis of coral status

When the sequence of sediment and disease-related conditions are examined across all colonies, only minor sediment presence (e.g., Fig. 4B) was observed on coral tissues (12 of 52 channel-side and four of 58 reference colonies) prior to June 2014. Major sediment accumulation including complete burial and several centimeter sediment berm (seemingly from colony expulsion of sediments, Fig. 4D) and the subsequent burial of colony edges was observed starting in early June 2014 (half of channel-side colonies compared to 1 of 58 reference colonies; Fig. S1). Bleaching was observed primarily in August–November 2014 (Fig. 4G; Fig. S1) with most of the colonies recovering (Fig. 4H). Most colonies of *Porites astreoides,* which occurred only at the reference reef, also were bleached in July 2015 at the end of the time series. The predominance of active disease signs and rapid complete colony mortality among channel-side colonies occurred between late November 2014 and late February 2015. However, most disease and rapid colony mortality among reference reef colonies occurred later (February–July 2015; Fig. S1).

Six channel-side colonies (11.5%) displayed complete or almost complete colony mortality directly associated with sediment burial (i.e., directly following in time and tissue regression over similar footprint as previously buried, Fig. S3). Although one reference colony

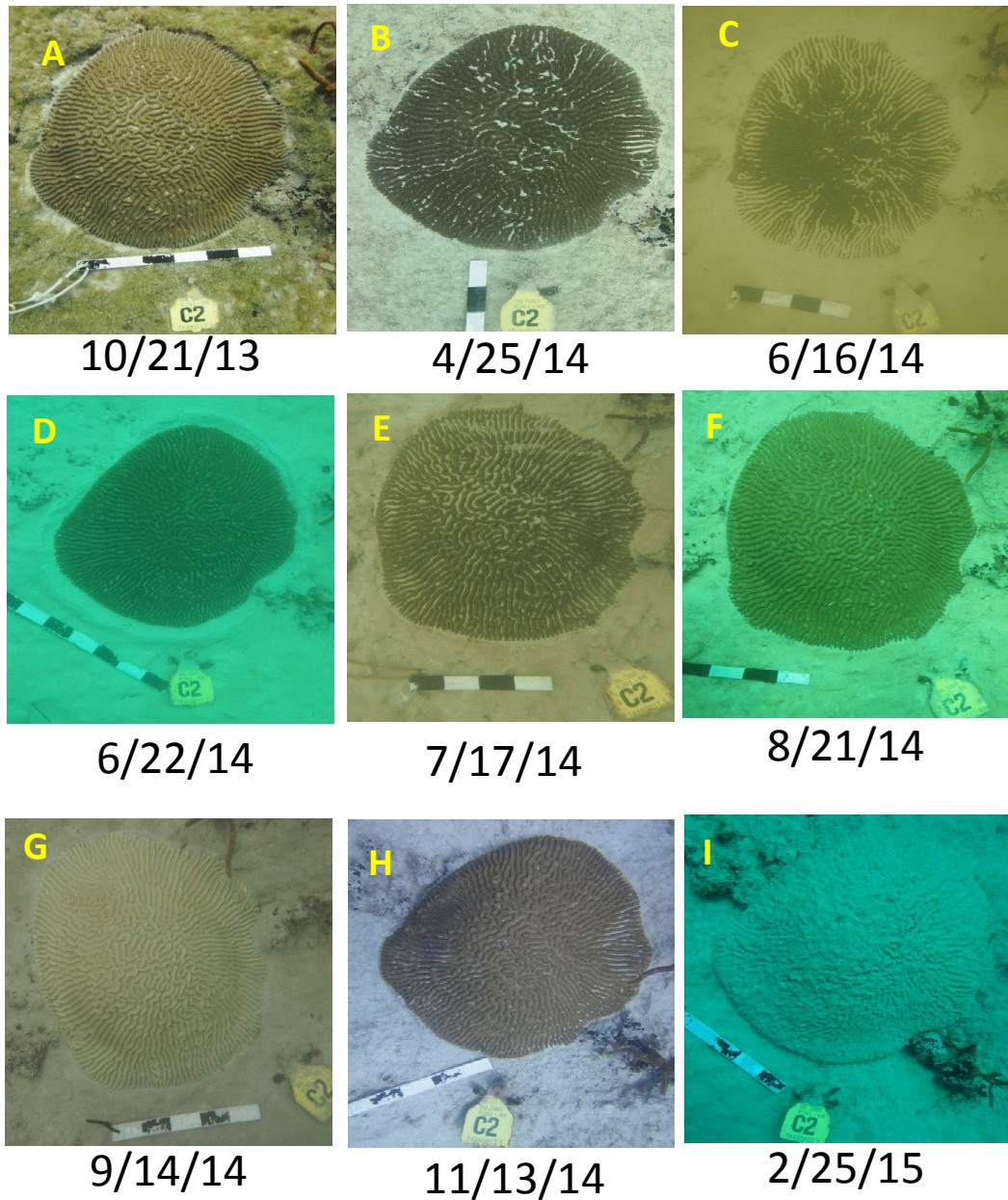

**Figure 4  Example coral time series.** Intermittent time series photos for a *Pseudodiploria strigosa* colony (designated R2N1 T1 C2) illustrating different conditions including sediment accumulation (B, E), partial burial (C), sediment 'berm' around coral margin (D), bleaching (G), recovery (H), and 'sudden death' (I). Also note the degree of accumulated sediment on the surrounding reef substrate. This colony was located in the Inner Reef north channel-side permit-monitoring site, within 30 m of the channel. Dates given as Month/Day/Year. Additional illustrations given in Fig. S3.

**Table 1 Partitioning of tagged colonies that experienced substantial sediment burial (complete or partial) and subsequent disease or death.** Risk is calculated as the percent of colonies in each category which manifest disease or death.

| Group | Sediment interaction | Subsequent disease/death | Risk |
|---|---|---|---|
| Channel-side 52 | Yes 26 | Yes 10 | 38% |
| | | No 16 | |
| | No 26 | Yes 9 | 34% |
| | | No 17 | |
| Reference 59 | Yes 1 | Yes 0 | 0% |
| | | No 1 | |
| | No 58 | Yes 9 | 15% |
| | | No 49 | |

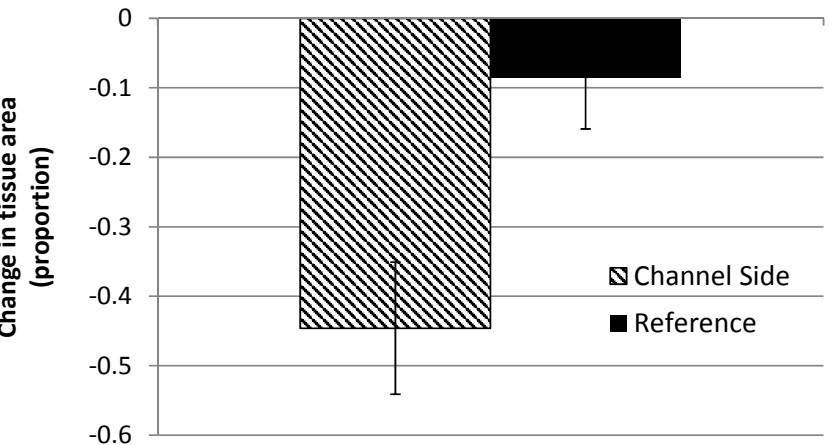

**Figure 5 Proportional change in coral tissue area (mean ±1 SE) for tagged colonies between the baseline and post-construction period (∼18 months).** $N = 55$ or 60 colonies (Channel-side, Reference, respectively). Channel-side colonies lost significantly more tissue.

appeared to experience a minor degree of partial sediment burial of its edges, it manifested only modest partial mortality (Fig. S3C). The occurrence of disease and complete colony mortality for reference colonies was less than half that observed for channel-side colonies (Table 1), although disease occurrence was similar between channel-side colonies that were observed with substantive partial burial (38%) versus those that were not (34%).

The tagged colonies at the channel-side location ($n = 55$), without regard to particular conditions or attributions of coral loss, showed over 4× greater tissue loss on average than the reference colonies ($n = 58$, Fig. 5). This includes 17/55 (31%) channel-side colonies versus 6/58 (10%) reference colonies which suffered complete colony mortality. Meanwhile, 48% of reference colonies displayed positive growth over the course of the project, compared with only 18% of channel-side colonies.

## DISCUSSION

A severe warm thermal stress (*Eakin et al., 2016*; *Manzello, 2015*) and coral bleaching event affected south Florida coral reefs beginning in autumn 2014 (MW Miller, pers. obs., 2014; also documented in regional bleaching surveys as 30–55% prevalence of bleaching in the sub-regions spanning Miami-Dade and Broward county; data available from Florida Reef Resilience Program at http://frrp.org/temp/JCDM3VBD/CoralDiseaseBySubregion.html; Fig. S1). As often occurs (*Muller et al., 2008*; *Miller et al., 2009*), the coral bleaching event was followed by severe but patchy coral disease outbreaks and mortality which were reported anecdotally throughout the region starting in winter 2014–2015 (Fig. S1). *Precht et al. (2016)* provide documentation of the origin, high lethality, and spread of this severe disease outbreak. These authors report the origination of this regional outbreak at the Inner Reef south reference site (referred to as 'Virginia Key', Fig. 1, located ~1.2 to 1.4 km south of the channel) in Sept. 2014 and the propagation of the outbreak both to the north and, more slowly, to the south.

Both bleaching and disease are documented in the time series observations of corals in both the Inner Reef north channel-side and reference populations (Fig. 4 and Fig. S3C ) examined in the present study. Despite these coincident disturbances, analysis of tagged coral colony condition during the course of the dredging project shows significant and large effects in terms of more severe coral tissue loss (almost 5×) and increased risk of disease and death (>double) in the immediate vicinity of the dredged channel, in comparison with project-chosen reference reef. The permit-mandated monitoring plan did not, however, incorporate spatial coverage of potentially impacted reef areas farther than 50 m from the channel that would aid in determining the spatial extent of impact. We implemented the post-hoc sampling (i.e., ~8 mos after dredging was completed) to partially address this gap. Although the determination of causes of coral mortality or partial mortality is always problematic, we compared the prevalence of several coral conditions and the persistent levels of standing sediment on reef substrates at a gradient of potential impact locations with the reference location to aid in delineation of the extent of sedimentation impact.

This post-hoc survey showed substantial differences between the assessment locations and the reference location in terms of standing sediment and coral condition. Using the most objective measures such as sediment depth (almost 10× at the 200 m location) and prevalence of recent partial mortality (~double across all assessment locations), significant contrast is evident with the reference location. This pattern also is consistent with the less-objective or ephemeral parameters in our survey, such as the attribution of partial mortality patterns as 'halos' or the presence of sediment on live coral tissue. Unfortunately, there are no directly comparable baseline data for these parameters, raising a potential concern that high sediment impact, relative to the reference area, may be characteristic of a reef area adjacent to a high-traffic channel, rather than a direct effect of dredging. Baseline sampling was conducted by USACE at a gradient of sites out to 450 m in the Inner Reef North sector in 2010 and recorded a binary score (yes/no) for the presence of an unspecified 'sedimentation' condition and the overall prevalence of coral partial mortality (not specified whether recent or not). The 'sedimentation' score was reported

as 'yes' at all sampled sites except the Inner Reef North site at 200 m where a 'no' was recorded for 'sedimentation'. The dive site for one of the two sampled in the current study at 200 m distance (200-RR, Table S1) was overlain with the 2010 transect location (as described in 'Methods') and showed 63% cover of SOHB and a mean sediment depth of 1.7 cm, apparently a very different condition than the absence of 'sedimentation' reported in 2010. Similarly, in 2010, the prevalence of coral partial mortality was reported as 3.1% and showed no significant relationship with distance from the channel (*US Army Corps of Engineers, 2011*). Our survey results record evidence of the severe impacts of regional coral stressors, such as thermal stress and disease in that the prevalence of recent partial mortality for reference area corals was 7× higher (21 ± 3.5 %, mean ± 1 SE) than the 2010 baseline assessment; however, the locations in the vicinity of the channel (up to 700 m distant) had values double those at the reference location (44 ± 3.4%, mean ±1 SE). This pattern is also consistent with the reported results of *Pollock et al. (2014)* showing that extended exposure to dredging project-related sediment plumes was a significant driver of increased occurrence of compromised conditions of reef corals.

Neither the dredging process nor the Port of Miami entrance channel environment was conducive to a simple sedimentation gradient leading away from the channel and into the coral reef habitat. While the material dredged from the federal channel was predominantly limestone and sand below a thin layer of silt, the proportion of silt within material from the inner harbor was generally higher than from the entrance channel. Three types of dredges (clamshell, hopper, and pipeline) operated at various times and locations, and the amount of sediment suspended from these dredges can differ substantially under normal operation (*McLellan et al., 1989*). In addition, the pipeline dredges occasionally used a process known as "roller chopping" (punching the substrate with the cutterhead to pre-treat and fracture rocky substrate) with the suction deactivated. A spider barge staged at various locations along the outer entrance channel collected dredged material from the pipeline dredges and distributed the material to hopper barges for transport to the ocean disposal site. The permit did not set controls for the overflow of sediment-laden effluent from this process, and the overflow of sediments may have been substantial and variable. Likely additional sources of sedimentation from dredging operations include the disposal doors in the hopper barges not sealing shut properly leading to leakage from the barges. Given the differences in sediment characteristics and the manner and rate of sediment influx to the water column coupled with the varying oceanographic conditions along the 8.9 km length of the project (including inner harbor work) during the many months of dredging, the lack of a simple linear sedimentation or impact gradient leading from the channel is not surprising.

While sediment movement and deposition is a normal process in a coral reef ecosystem, offshore coral reefs are not capable of developing or sustaining ecological functions when substrates are covered by sediment over prolonged periods. The presence of deep sediment pockets within patchy reef habitats may also be a normal reef habitat feature. However, the presence of emergent sessile invertebrates (particularly soft corals, but also scleractinian corals and sponges, Figs. 2C–2D, Fig. S2) in much of the area of observed deep sediment in our post-hoc surveys clearly indicated recent, extreme levels of accumulation and implies that additional, uncountable scleractinian corals have been buried in these areas.

Although our replication was not adequate to discern habitat effects, the post-hoc survey results suggest differences in the severity of sedimentation impact between the habitat types. Most survey locations in the vicinity of the channel showed a trend for higher sediment cover and depth in the Linear Reef than the Ridge-shallow habitat (Table S1). The Linear Reef habitat 200 m north of the channel appears to have been the most severely impacted as this location had the highest cover (43%) characterized as DSOHB (4.0 cm or greater sediment over reef; Table S1), the highest measured maximum sediment depth (10.0 cm), and the highest prevalence of sediment halo (26%; Table S1). It is possible the prevalences of recent mortality and sediment accumulation at Linear Reef (200 m) are underrepresented, when compared to other sites, because many low-lying scleractinian colonies have been completely buried.

Sedimentation on reefs can reduce coral recruitment, survival, and settlement of coral larvae (*Fabricius, 2005*; *Erftemeijer et al., 2012b*; *Jones, Ricardo & Negri, 2015*) and suppress colony growth (*Bak, 1978*). Our study focused on the reef sector which experienced the greatest duration of sediment plumes during the dredging project and relies heavily on the representativeness of the reference reef. This reference area was chosen to provide a representative comparison, comprising similar reef habitats, prior to initiation of dredging. Coral disease impacts can be very site specific (e.g., *Miller et al., 2014*), so a more spatially comprehensive quantitative analysis of coral disease effects both in potential impact areas and regionally would be beneficial. However, the increased prevalence of indicators of sedimentation stress and partial mortality, as well as persistent standing sediment on reef substrates at the Port of Miami sedimentation assessment areas (Figs. 2 and 3), all suggest the cumulative sedimentation was much greater across the assessment sites, when compared to the reference area, and mortality and loss of function of reef organisms resulted. When considering the findings of this study coupled with the findings of *Barnes et al. (2015)*, sediment plumes and deposition via multiple pathways derived from dredging activities at Port of Miami are the most plausible drivers for this pattern.

The implementation of seasonal shutdowns for dredging projects near coral reefs has largely been based on protecting corals during major spawning events. Unfavorable conditions during a coral spawning period could negate the entire reproductive output for the year (*Harrison et al., 1984*). Conducting dredging activities at appropriate times to avoid spawning periods would constitute a best management practice (*Jones, Ricardo & Negri, 2015*). Recommendations for reduced or halted dredging range from one week based on the known coral spawning period in Singapore (*Erftemeijer et al., 2012b*) to as many as five months per year based on known spawning periods in northern Western Australia (*Baird et al., 2011*). This best management practice could provide enhanced protection if shutdowns were also to coincide with increasingly predictable seasonal thermal stress events (*Van Hooidonk et al., 2014*; *Manzello, 2015*) or less predictable cold water stress events. However, fixed predictable shutdowns are likely more practical especially if they are included in dredge contract bid specifications. In addition, feedback monitoring to effectively execute adaptive management may need to also include light attenuation and suspended sediment concentration. In Florida (USA), this practice has not been well-socialized in the regulatory context for coral protection with the exception of the Key West Harbor Dredging Project,

where the contract provided for limited relocation of the dredge when coral health and sediment accumulation levels exceeded allowable thresholds (*US Navy, 2003*).

Another port expansion at Port Everglades, located approximately 37 km north of Port of Miami, is on the horizon for southeast Florida. The construction plans at Port Everglades are similar in scale with USACE proposing to remove 4.21 million $m^3$ of material. However, recent thermal stress and disease impacts have rendered the baseline reef condition as further impaired and less able to tolerate increments of 'standard' sedimentation stress associated with dredging activities in the past (e.g., *Marzalek, 1982*). The proposed Port Everglades monitoring plan is similar to that used for Port of Miami (*US Army Corps of Engineers, 2015*), though expected to be modified to capture lessons learned in Miami. Notable improvements to the monitoring plan would include monitoring standing sediment depth, sediment-associated stressors (e.g., coral halo), light attenuation, suspended sediment concentration, near-realtime information feedback on monitoring outcomes, observations from other parties, and attending to regional disturbances such as warm-water and coral disease events. It is also crucial that the monitoring be undertaken at a spatial scale that can capture the indirect effect area (e.g., 3 km based on *Fisher et al., 2015*) with near realtime status/extent of sediment plumes via remote sensing (e.g., *Barnes et al., 2015*), and local oceanographic conditions as important guides. Time series analysis of permanently marked corals could be used in concert with continuous water quality monitoring and sediment depth measurements to make use of, and possibly contribute to, a rapidly evolving peer-reviewed literature concerning thresholds for environmental impacts of dredging projects (e.g., *Jones et al., 2016*; *Nelson et al., 2016*; *Fisher et al., 2015*). Such an approach could provide an early predictor of when and where sedimentation impacts are occurring to adaptively manage the dredging. Inclusion of this type of monitoring could help in the development of no-work windows, including when regional thermal events are ongoing. Even if no-work windows or seasonal shutdowns are not implemented, monitoring thresholds could still be identified to serve as a warning that coral impacts will exceed what was predicted under normal conditions. If lessons learned from Port of Miami expansion are not memorialized, well-socialized, and fully realized in the monitoring requirements developed for future projects, such as the Port Everglades expansion, avoidable impacts may recur.

## ACKNOWLEDGEMENTS

Field or logistical support was provided by J Javech, J Europe, R Pausch, J Blondeau, S Meehan and Callaway Marine Technologies. GIS support provided by K Hanson.

### Funding

This work was funded by the NMFS Southeast Regional Office, NMFS Restoration Center, and NOAA Coral Reef Conservation Program. The funders had no role in study design, data collection and analysis, decision to publish, or preparation of the manuscript.

## Grant Disclosures

The following grant information was disclosed by the authors:

NMFS Southeast Regional Office.

NMFS Restoration Center.

NOAA Coral Reef Conservation Program.

## Competing Interests

The authors declare there are no competing interests.

## Author Contributions

- Margaret W. Miller conceived and designed the experiments, performed the experiments, analyzed the data, wrote the paper, prepared figures and/or tables, reviewed drafts of the paper.
- Jocelyn Karazsia conceived and designed the experiments, analyzed the data, wrote the paper, prepared figures and/or tables, reviewed drafts of the paper.
- Carolyn E. Groves analyzed the data, prepared figures and/or tables, reviewed drafts of the paper.
- Sean Griffin and Tom Moore conceived and designed the experiments, performed the experiments, reviewed drafts of the paper.
- Pace Wilber conceived and designed the experiments, reviewed drafts of the paper.
- Kurtis Gregg performed the experiments, reviewed drafts of the paper.

## Data Availability

The raw data has been supplied in Data S1.

## Supplemental Information

Supplemental information for this article can be found online at http://dx.doi.org/10.7717/peerj.2711#supplemental-information.

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
