# Peer review of "Detecting sedimentation impacts to coral reefs resulting from dredging the Port of Miami, Florida USA"

_PeerJ, doi:10.7717/peerj.2711_

## Round 0.1 · original submission · Minor Revisions

The paper will be acceptable after minor revisions to address the reviewer's comments.

Please note that Reviewer 2's review is in an attached document.

Reviewer 1 ·

Basic reporting

adequate

Experimental design

adequate with caveats mentioned in general comments.

Validity of the findings

I believe the findings are valid. The authors have done a good job with available and new information to sort out dredging effects and effects that have exacerbated.

Additional comments

Abstract is vague as to the purpose of the paper. The purpose of the paper is not clearly stated.

Why is this phrase “While the precise effects of the dredging on surrounding coral reefs are not well quantified” needed? Does this mean that your effects assessment is not well quantified?

Abstract says: “A regional warm-water mass bleaching event followed by a coral disease outbreak during this same time frame confounded the assessment of dredging-related impacts to coral reefs adjacent to the federal channel.” I’m not sure confounded which means “bewildered” is the right word. It made the assessment more difficult, but it still was possible.

Abstract says: “Results indicate increased sediment accumulation… occurred within coral reef sites located closer to the channel.” This seems to be much less of a result/conclusion possible, given the long discussion in the paper about how the sedimentation effects extended over 700 m from the channel as opposed to USACE monitoring very close to the channel. Yes they might have been most intense near the channel, but it is important to note that the effects extended 10x farther than the EIS assessment took into account. The discussion of about monitoring to identify sources and the suggestion not to dredge during the summer are discussion points.


P3, Fig 1. Fig 1 only vaguely shows the reef zones (features) you mention of nearshore ridge, Inner and outer reefs. Since the zones appear important to identify, it would be better to show a zone map of equal size and scale next to the air photograph which more clearly identifies. The enlargements of the air photograph currently on the right of Fig 1 could be another figure.

P4 top. HF radar studies by Shay of University of Miami indicate the spinoff eddies are quite common. Consider citing.

P4 “included conditions for biological monitoring areas adjacent to the channel along each of six coral reef or hardbottom features” I guess you mean the 6 “features” are the nearshore ridge, Inner Reef, and Outer Reef, both north and south?

P4, lines 94-100. Be consistent in use of stations and sites.. I’m assuming these mean the same thing.

Lines 101-103: “partitioned natural drivers of sediment plumes in the vicinity of the Port of Miami channel from dredging associated sediment plumes.” Not clear what this means.

Was the purpose of the study to evaluate if the USACE EIS plan of only looking at the nearshore within 70 m of the channel was adequate? If so, you should state this explicitly.

I think 111-118 should be written more clearly and come sooner to present the rationale and purpose of the paper.

P5, lines 112-115. Confusing. Were there 5 sediment “locations” at the reference area? Figure 1 shows 5 sediment “locations” at the Inner Reef and 4 at the Ridge Shallow. Did you not looks at those of the Ridge Shallow? Figure 1 shows no “locations” circles at the reference site. Why?

Later in 126-127, you say: “transects were evenly distributed in both Ridge-shallow (RR) and Linear Reef (LR) habitat types”. There are 4 dots of sediment “locations” in Ridge Shallow. Were these used? Confusing

P5 Line 129. First time you have introduced Ridge Shallow and Linear Reef terms. They are not previously identified or shown in Fig 1.

p. 6, line 138. Confusing. How many sediment locations, dive sites per location, and habitats. Needs to be clearly presented. There are three exceptions given which adds to the confusion.

Lines 141-142. Describe the 2 transects.

151-152. Did the belt transects include the line transects? Unclear

164. Describe how you did the “ranks”.

166. So you pooled the transect data from each of the 2 sites at a given distance. How pooled? You considered you had 4 transects instead of 2 at each distance?

172. It does not appear you mentioned transects at the channel sides. Are those USACE transects or yours? Are they the 100m transects? Where are the channel side transects? What are they for? Are these located within 70 m in the three yellow and three red rectangles in the left insert of Fig 1 for the Inner Reef and Outer Reef North and South? Are these belt transects?

208. Confusing since you have not explained ridge-shallow and linear reef previously.

254-265. Was there a difference in partial and total mortality between tagged channel side corals and reference corals?

268-281. This section appears to me to be a result and not a discussion.

378 For threshold assessment, consider also referring to:
Proceedings of the 11th International Coral Reef Symposium, Ft. Lauderdale, Florida, 7-11 July 2008, REAL-TIME CORAL STRESS OBSERVATIONS BEFORE, DURING, AND AFTER BEACH NOURISHMENT DREDGING OFFSHORE SOUTHEAST FLORIDA, USA. L. Fisher, K. Banks, D. Gilliam, R. E. Dodge, D. Stout, B. Vargas-Angel, Brian K. Walker

Annotated reviews are not available for download in order to protect the identity of reviewers who chose to remain anonymous.

Reviewer 2 ·

Basic reporting

see attached review

Experimental design

see attached review

Validity of the findings

see attached review

Annotated reviews are not available for download in order to protect the identity of reviewers who chose to remain anonymous.

---

## Round 0.2 · accepted · Accept

You have adequately responded to the reviewer's suggestions and the paper is now ready for publication